# Active Learning Principles for In-Context Learning with Large Language Models

**Katerina Margatina**◇∗ **Timo Schick**† **Nikolaos Aletras**◇ **Jane Dwivedi-Yu**†
◇University of Sheffield †FAIR, Meta
{k.margatina, n.aletras}@sheffield.ac.uk
janeyu@meta.com

## Abstract

The remarkable advancements in large language models (LLMs) have significantly enhanced predictive performance in few-shot learning settings. By using only a small number of labeled examples, referred to as demonstrations, LLMs can effectively perform the task at hand through in-context learning. However, the process of selecting demonstrations for maximizing performance has received limited attention in prior work. This paper addresses the issue of identifying the most informative demonstrations for few-shot learning by approaching it as a pool-based Active Learning (AL) problem over a single iteration. We compare standard AL algorithms based on uncertainty, diversity, and similarity, and consistently observe that the latter outperforms all other methods, including random sampling. Our extensive experimentation involving a diverse range of GPT and OPT models across 24 classification and multi-choice tasks, coupled with thorough analysis, unambiguously demonstrates the importance of using demonstrations that are semantically similar to the domain of the test examples. In fact, we show higher average classification performance using "similar" demonstrations with GPT-2 (124M) than random demonstrations with GPT-Neox (20B). Notably, while diversity sampling shows promise, uncertainty sampling, despite its success in conventional supervised learning AL scenarios, performs poorly in in-context learning.

## 1 Introduction

The field of Natural Language Processing (NLP) has recently witnessed a remarkable paradigm shift with the emergence of in-context learning with large language models (LLMs), also referred to as few-shot learning (Brown et al., 2020). Traditionally, NLP systems heavily relied on supervised learning approaches, where large amounts of labeled training data were necessary to achieve high

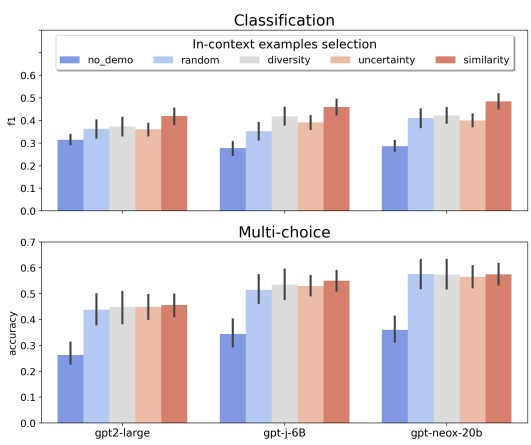

Figure 1: Performance of different in-context selection algorithms in classification and multi-choice tasks.

predictive performance. However, in-context learning has changed this status-quo by enabling LLMs to learn from limited, context-specific examples and adapt to new tasks and domains with remarkable proficiency (Zhao et al., 2021; Chowdhery et al., 2022; García et al., 2023; Wei et al., 2023b; Touvron et al., 2023; Bubeck et al., 2023). Unlike more traditional approaches, which require extensive retraining or fine-tuning for every new task, in-context learning empowers LLMs to generalize from a few examples that are fed to the model through prompting to learn a new task at hand, without any weight updates.

The data efficiency of few-shot in-context learning of LLMs is indeed remarkable with only a small number of demonstrations.[1] Still, such demonstrations constitute *labeled* data examples, raising two key questions: (1) When faced with tasks where there is only *unlabeled* data available, how can we select the most appropriate samples to label and then use as in-context demonstrations? (2) When we have *labeled* data for a given task, how can

---

∗ Work done during an internship at FAIR, Meta.

[1]We use the terms *in-context examples*, *few-shot examples*, *demonstrations*, *descriptors* and *exemplars* interchangeably throughout the paper.

we efficiently identify the most informative combination of demonstrations for in-context learning? Answering these questions is essential to ensure effective and efficient few-shot learning using LLMs.

A growing line of work has investigated how in-context learning works (Reynolds and McDonell, 2021; Razeghi et al., 2022; Xie et al., 2022; Ye et al., 2023b), which demonstrations to use (Liu et al., 2022; Zhang et al., 2022b; Wu et al., 2022; Kim et al., 2022), how to form the prompt (Zhao et al., 2021; Lu et al., 2022; Yang et al., 2023) and whether ground truth labels matter (Webson and Pavlick, 2022; Min et al., 2022; Yoo et al., 2022; Wang et al., 2022; Wei et al., 2023b). Still, to the best of our knowledge, no prior work has explored the problem of in-context demonstration selection explicitly through the lens of active learning (AL).

Based on the core principle that not all data points are equally useful, AL (Cohn et al., 1996; Settles, 2009) aims to identify the most informative instances from a pool of unlabeled data for annotation. Iterating through model training, data acquisition and human annotation, the goal is to achieve data efficiency. A data-efficient AL algorithm ensures that a model achieves satisfactory performance on a withheld test set by selecting only a small fraction of the unlabeled data for annotation that typically is better than randomly selecting and annotating data of equal size.

In this paper, our main aim is to redefine the concept of data efficiency within the framework of in-context learning inspired by conventional active learning settings. For this purpose, we assume that given a pool of labeled or unlabeled data, the objective is to identify a set of $k$ examples that will serve as demonstrations to an LLM, resulting in optimal performance on a held-out test set. Given this formulation of data efficiency, we explore the effectiveness of the most prevalent AL approaches based on uncertainty (Lewis and Gale, 1994; Cohn et al., 1996; Gal et al., 2017), diversity (Brinker, 2003; Bodó et al., 2011; Sener and Savarese, 2018) and similarity (Margatina et al., 2021; Kirsch et al., 2021; Liu et al., 2022), as demonstration selection methods for in-context learning (Figure 1).

Our key contributions are as follows:

- We formulate the selection of in-context examples as a single iteration AL problem and explore the effectiveness of four standard approaches: *uncertainty*, *diversity*, *similarity* and *random* sampling.

- We evaluate 15 models, between 125M and 30B parameters, from the GPT (Radford et al., 2019; Brown et al., 2020; Black et al., 2022) and OPT (Zhang et al., 2022a) families in 15 classification and 9 multi-choice tasks, using different AL sampling techniques to select demonstrations for few-shot learning.

- We demonstrate that while diversity and uncertainty sampling perform slightly better than random sampling, choosing in-context examples that are semantically similar to the input test examples outperforms consistently all other methods by a large margin across model families and sizes in all tasks.

- We show that while uncertainty sampling is one of the strongest AL approaches in supervised learning, this does not generalize to in-context learning, where interestingly it underperforms. Our analysis, however, shows that larger models might perform better with uncertain demonstrations, hinting that uncertainty might be an emerging LLM ability.

## 2 Active In-context Learning

### 2.1 Problem Formulation

To build our in-context learning framework with actively acquired demonstrations, depicted in Figure 2, we borrow the formulation from the standard pool-based active learning paradigm. We consider an AL setting where we have a large pool of unlabeled data from which we want to sample a batch of $k$ data points using a data acquisition algorithm. We assume that these $k$ are subsequently labeled by humans (Figure 2, top). Instead of following the standard approach that involves multiple iterations of data selection and model training, we only perform a single iteration (Longpre et al., 2022), since we do not train or perform any model-in-the-loop updates. We use the acquired set of $k$ examples as demonstrations for in-context learning with an LLM (i.e., as part of the prompt). We assume the existing datasets as the pool from which to select these $k$ examples. The goal is to find the most informative examples from the pool, which are expected to yield improved performance on the test set when employed as a few-shot prompt, compared to demonstrations randomly sampled from the same pool. The resulting prompt consists of the concatenation of the $k$ acquired examples (text

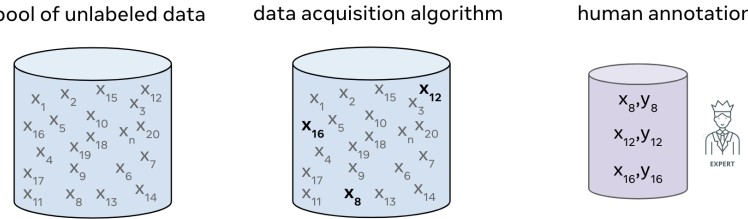

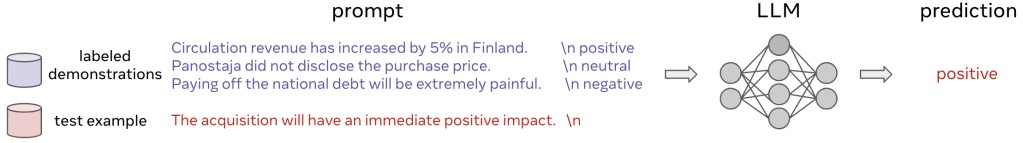

Figure 2: Top: Active data collection (single iteration). Bottom: Prompt construction and model inference.

inputs and labels with standard verbalizers), alongside the test example, repeated for all data instances in the test set (Figure 2, bottom).

## 2.2 Few-shot Data Acquisition Algorithms

We build few-shot data acquisition algorithms inspired by the most prevalent AL algorithmic families that are uncertainty sampling, diversity sampling and similarity (also known as test-aware sampling) (Zhang et al., 2022c). We acknowledge that there are more elaborate demonstration selection methods for in-context learning that are not considered in our experiments, such as Q-learning (Zhang et al., 2022b), Self Adaptive (Wu et al., 2022), SG-ICL (Kim et al., 2022), MI (Sorensen et al., 2022), *inter alia*. These methods fall beyond the scope of our analysis, as our objective is to gain insights into AL principles for in-context learning, rather than benchmarking all available demonstration sampling algorithms. Additionally, there are techniques, complementary to the aforementioned few-shot data selection methods, such as calibration (Zhao et al., 2021) and prompt re-ordering (Lu et al., 2022), which can further enhance few-shot learning performance, while also being out of the scope of our work.

**Random** The overarching objective of any data selection method, like AL algorithms, is to identify data points that, however used, yield superior models compared to randomly sampled data from the same pool which we consider as a baseline method.

**Diversity** The first data selection method that we use as a representative for the diversity family of methods is a simple clustering technique, similar

to Yu et al. (2022). Specifically, we first encode all data points in the pool of unlabeled data with Sentence-BERT (Reimers and Gurevych, 2019) embeddings and then we perform k-means clustering.[2] We choose the number of clusters to be $k$ and select one data point from each cluster. The underlying principle of this approach is that leveraging a diverse set of in-context examples can offer greater advantages compared to random sampling. This selection strategy ensures that the chosen demonstrations are likely to encompass a broad range of information, enhancing the overall effectiveness of the learning process.

**Uncertainty** The second approach is an uncertainty-based sampling algorithm that is based on SPELL, proposed by Gonen et al. (2022). Since we use an off-the-shelf LLM that does not have a fine-tuned classification layer, we cannot compute the model probabilities associated with each class (for a classification or multi-choice task). This essentially means that we cannot use standard AL uncertainty baselines such as maximum entropy or least confidence. Instead, we can use the loss, i.e., perplexity, of the LLM to score each candidate example from the pool. Gonen et al. (2022) define perplexity of the prompt as the perplexity of the full prompt sequence, including the input itself, and without the label, averaged over $1,000$ examples. Our approach is different since we want to evaluate the perplexity of each in-context example individually. We also do not do the averaging over a thousand examples as we wanted to make the method more general, without

---

[2]We use the implementation from https://www.sbert.net/examples/applications/clustering/.

the need to assume access to that many examples. The underlying principle guiding this approach is the belief that a high perplexity set of in-context examples can yield greater advantages compared to randomly sampling from the dataset (or at least for data efficiency in a supervised learning setting this is proven to enhance the learning process).

**Similarity**  Finally, the third AL algorithm we consider is based on KATE a kNN-augmented in-context example selection method proposed by Liu et al. (2022). This method retrieves examples from the pool that are semantically-similar to a test query sample. We use Sentence-BERT (Reimers and Gurevych, 2019) representations of both the pool and the test set to find the k-nearest neighbours. The rationale behind this approach is that the most similar demonstrations to the test example will best help the model answer the query. We have to highlight, however, that by definition each test example will have a different prompt, as the $k$ most similar demonstrations will be different. This is a crucial limitation of this approach compared to the others, as it assumes that we are able to acquire labels for any in-context example selected from the pool.

## 3   Experimental Setup

**Models**  We evaluate 15 LLMs in total, 8 models from the GPT (Radford et al., 2019; Brown et al., 2020; Black et al., 2022) and 7 from the OPT (Zhang et al., 2022a) family. We choose our models to span from a few million to tens of billions parameters, as we want to study how the model size affects the effectiveness of in-context example selection methods. All models considered in this work are publicly available.

**Tasks & Datasets**  Following Min et al. (2022), we evaluate all LLMs in 15 classification and 9 multi-choice tasks taken from the Crossfit (Ye et al., 2021) benchmark. We provide details for all tasks and datasets considered in the Appendix A.1.

**In-context Learning Prompting**  Unless specified otherwise, we sample $k$=16 demonstrations, i.e., labeled data, from the pool with each AL method. After collecting the $k$ input-label pairs, we concatenate them all together with the test example that we want to make a prediction for to form the LLM prompt (Figure 2). Our implementation, including prompt verbalizers, is based on those by Min et al. (2022) and Yoo et al. (2022).

## 4   Results

Figure 3 shows the results on few-shot in-context learning across all data acquisition methods (random, diversity, uncertainty and similarity), model families (GPT and OPT) and tasks (classification and multi-choice question answering).[3] Overall, we observe the anticipated trend of performance enhancement with increasing scale, particularly notable in the multi-choice tasks for both OPT and GPT models.

Still, the most remarkable finding is the substantial performance improvement achieved by selecting similar in-context examples for few-shot learning, particularly in classification tasks. This observation aligns with the findings reported by Liu et al. (2022), who demonstrated similar patterns in sentiment analysis tasks with GPT-3. Our results indicate that the selection of appropriate demonstrations can hold greater significance than the number of model parameters, at least within the scope of the models evaluated in this study. In multi-choice tasks, similarity is also the top-performing acquisition method, while the other three approaches exhibit closely competitive performance.

The data selection method based on diversity is consistently the second best approach after similarity (with very few exceptions in the multi-choice tasks for OPT models). Even though it is not the top performing method, we can consider that consistently outperforming random sampling is a strong signal that diversity in the demonstrations is a characteristic of effective demonstrations. Levy et al. (2022) explore the setting of compositional generalization, where models are tested on outputs with structures that are absent from the training set and thus selecting similar demonstrations is insufficient. They show that combining diverse demonstrations with in-context learning substantially improves performance for the task of compositional generalization semantic parsing.

Remarkably, uncertainty sampling, typically regarded as one of the best approaches for traditional supervised AL (Shen et al., 2017; Margatina et al., 2022; Schröder et al., 2023), exhibits the lowest performance. This finding contradicts the conventional AL principles that suggest selecting a few highly uncertain labeled data points for data efficiency. Similar to our findings, Gonen et al. (2022) explore the performance variabilty of dif-

---

[3]We provide the results per dataset and model in the Appendix A.2, including the majority vote baseline.

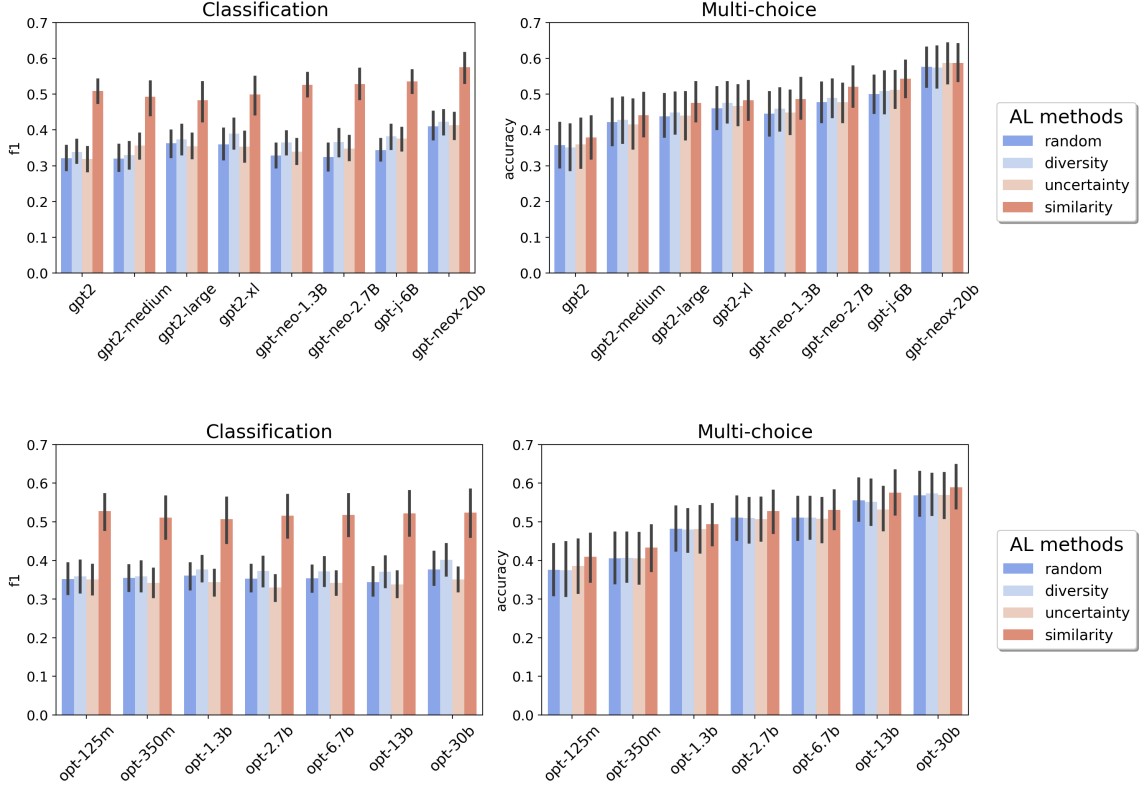

Figure 3: Results for various GPT (top) and OPT (bottom) models and AL methods averaged over 15 classification and 9 multi-choice tasks. *Similarity* is consistently the best performing approach overall, followed by *diversity* and *random*. Interestingly, we observe that *uncertainty* sampling underperforms in this setting of in-context learning.

ferent prompts (consisting of randomly sampled demonstrations) for in-context learning using uncertainty, and find that the lower the perplexity of the prompt is, the better the prompt is able to perform the task. Still, in a later analysis we show that larger models might be able to handle high uncertain prompts better than the smaller ones (§5.4).

## 5 Analysis

### 5.1 Effect of Model Size

In order to gain some intuition on the effect of scale, we group together GPT and OPT models with similar number of parameters. We provide the results in Figure 4. Even after aggregating the results from both model families, we do not see any specific pattern as the model parameters increase. We wanted to explore whether the largest models of our collection would behave differently under the varying in-context learning settings, thus perhaps attributing such a behaviour to potential emergent abilities of the bigger LLMs, but we observe the same patterns (in terms of ranking between the considered data selection methods). We believe that this is an interesting avenue of future research,

especially as models grow and, most likely, will continue to grow exponentially in terms of model parameters. Our findings show that the in-context learning ability of models from a few millions to a few billions of parameters follows similar patterns. However, this might not be the case when studying even larger models, as primary results hint (Rae et al., 2022; Wei et al., 2023b; Chowdhery et al., 2022; Touvron et al., 2023).

### 5.2 Ground Truth Demonstrations

We next delve into the debate of whether ground truth demonstrations, i.e., providing the correct label to the in-context examples, is crucial for high performing in-context learning. Various findings have shown mixed results for randomly sampled data, which essentially means that the benefit of ground truth labels depends on the label space or the distribution of inputs specified by the demonstrations (Min et al., 2022; Yoo et al., 2022). In our analysis, we differentiate from prior work by exploring the importance of ground truth demonstrations in the case of leveraging similar in-context examples (§2.2). The rationale is that if the find-

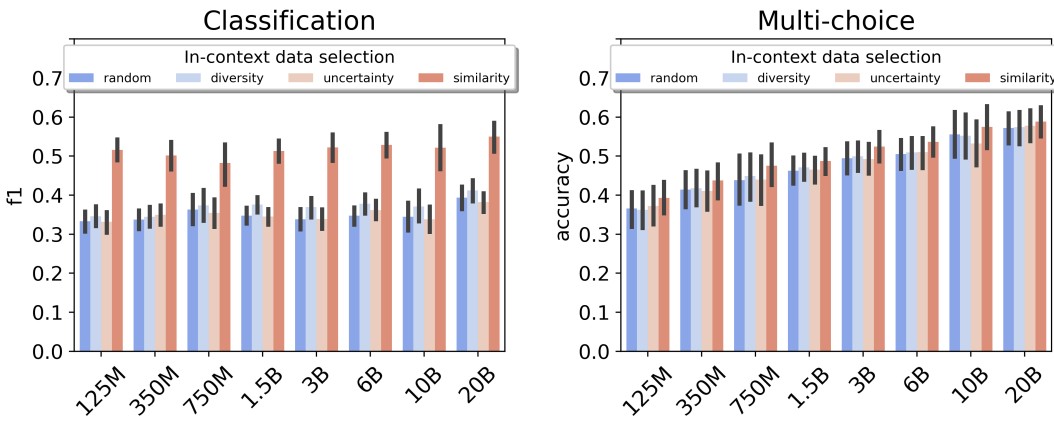

Figure 4: Results per model size.

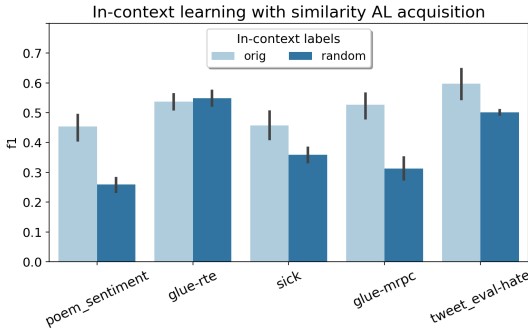

Figure 5: Effect of ground truth labels on in-context learning with with the similarity AL selection method.

ings of Min et al. (2022) ubiquitously hold, then the performance should only marginally drop if we replace ground truth labels with random ones. If the high performance of the `similarity` acquisition method can be retained, we would be able to construct an efficient and effective in-context selection algorithm that would be agnostic to correct labels. However, we find that this is not the case. We show in Figure 5 that for almost all datasets considered in this part of analysis, the performance with random labels drops significantly as expected. There are cases where replacing the original labels with random ones as in Min et al. (2022) retains the same performance (e.g., in the `glue-rte` dataset), but this is certainly a finding that does not generalize overall. In summary, we find that ground truth demonstrations are crucial for high performing, robust in-context learning (Yoo et al., 2022).

### 5.3 Most vs. Least Similar Demonstrations

To investigate the striking effectiveness of the *similarity*-based acquisition strategy, we conduct additional experiments where we invert the approach

and choose the *least* similar examples from the pool to form the prompt. This investigation aims to ascertain whether the remarkable performance gains can be attributed solely to the semantic similarity between the demonstrations and the test input. The results depicted in Figure 6 substantiate our hypothesis, demonstrating a significant performance drop when employing opposite examples from the pool as in-context exemplars. While this pattern is particularly pronounced in the classification tasks, it consistently emerges across different model sizes and task types. Hence, we can assert that *maximizing semantic similarity between the demonstations and the input test sample* is an unequivocally vital attribute for achieving successful in-context learning outcomes with LLMs. Future endeavors in the field of building effective in-context learning frameworks should incorporate this principle to enable data-efficient algorithms that can fully harness the potential of LLMs.

### 5.4 Most vs. Least Uncertain Demonstrations

Along these lines, we also opt to examine the duality between selecting the most or the least uncertain in-context examples from the pool. We show the results of these experiments for the GPT models in Figure 7. Interestingly, we observe that while the smaller language models (`gpt2`, `gpt2-medium`, `gpt-large`) perform better with the least uncertain prompts, the larger models seem to start benefiting from the demonstrations with high uncertainty. This is particularly clear in the largest model of our collection, `GPT-Neox` (20B parameters). This interesting finding shows that even larger models will most likely perform better with high entropy in-context examples, similar to their supervised learn-

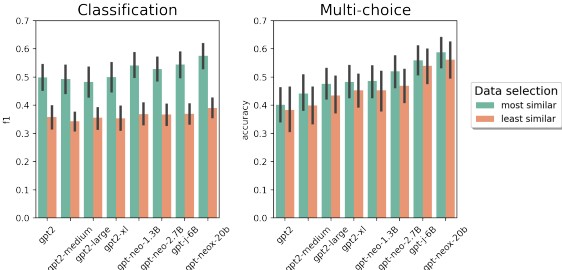

Figure 6: Most vs. least similar in-context examples.

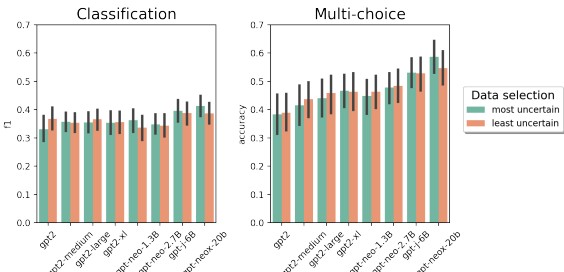

Figure 7: Most vs. least uncertain in-context examples.

ing counterparts. Such findings open a plethora of research questions regarding understanding how in-context learning works (Reynolds and McDonell, 2021; Razeghi et al., 2022; Xie et al., 2022; Min et al., 2022), how AL and data acquisition methods reshape with larger language models or whether we can properly investigate potential emergent abilities of LLMs acquired by model scaling (Wei et al., 2022; Schaeffer et al., 2023).

## 5.5 Evaluation with Different Metrics

Finally, we want to provide a clear overview of our experiments and summary of our findings, while making some clarifications regarding how we evaluate and compare different approaches to in-context learning. Figure 8 shows the results for in-context learning with random sampling, three data selection techniques inspired by AL (§2.2), namely `diversity`, `uncertainty` and `similarity`, and a zero-shot baseline where no labeled examples are included in the prompt (`no_demo`). We show that in-context learning with $k$=16 demonstrations consistently outperform zero-shot learning for an average of 15 classification tasks for `gpt2-large`, `gpt-j` and `gpt-neox`. Next, we observe that the best performing in-context example selection method is by a clear margin `similarity`, followed by `diversity`. This finding corroborates the original hypothesis of AL that, indeed, *not all data is equal* and there exist *more informative* data subsets

in the pool that can be used as in-context exemplars. We can see that the `uncertainty` baseline, which is usually top performing in supervised AL, generally underperforms in the few-shot setting. Still, there is some evidence that this could change with even larger and better models (§5.4). Finally, delving into the debate on whether ground truth labels matter or not (Min et al., 2022; Yoo et al., 2022), we show that replacing original with random in-context labels hurt significantly the performance of `similarity`, the best data selection method (§5.2).

We further emphasize the significance of employing a meticulous evaluation framework, particularly in the selection of appropriate metrics. In Figure 8, we illustrate the same classification experiments, but with the $F_1$ score plotted on the left and accuracy on the right. The use of $F_1$, the conventional metric for classification tasks, reveals a distinct ranking among the various AL methods, with `similarity` exhibiting the best performance, followed by `diversity`. Conversely, when employing accuracy to compare the methods, `diversity` emerges as the top approach, followed by `similarity` and random selection. This disparity highlights the potential for misconceptions or obscured findings, underscoring the need for caution when evaluating and comparing different methods across various models within the in-context learning framework (Dehghani et al., 2021; Min et al., 2022; Yoo et al., 2022; Tedeschi et al., 2023).

## 6 Related Work

### 6.1 Understanding In-Context Learning

Few-shot in-context learning with LLMs has garnered significant attention in recent NLP research. Simply concatenating a few labeled examples to form the prompt for the LLM results in high performance gains, even outperforming fine-tuned models (Brown et al., 2020; Chung et al., 2022; Ouyang et al., 2022; Dong et al., 2022). This has naturally lead to study its effectiveness with multiple few-shot learning benchmarks such as `Crossfit` (Ye et al., 2021) and `BigBench` (Srivastava et al., 2022).

Another active area of research is on understanding how in-context learning works (Xie et al., 2022; Garg et al., 2022; Akyürek et al., 2022; Xie et al., 2022; Pan et al., 2023), and what are its strengths and limitations (Webson and Pavlick, 2022; Jang et al., 2022; Levy et al., 2022; Shi et al., 2022; Agrawal et al., 2022; Wei et al., 2023b; Ye et al., 2023b). Previous work has explored the effec-

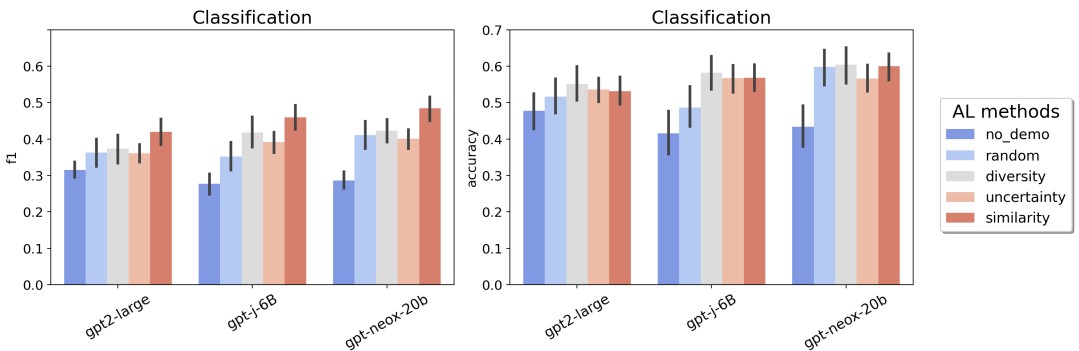

Figure 8: The ranking of data selection methods is different depending on the metric used.

tiveness of the chain-of-thought prompting technique (Wei et al., 2023a; Wang et al., 2022; Madaan and Yazdanbakhsh, 2022), while other studies try to determine the importance of in-context ground truth labels, with Min et al. (2022) showing that random labels do not hurt performance considerably and Yoo et al. (2022) providing a rebuttal. Wei et al. (2023b) explain that model size plays an role in the effect of ground truth labels, showing that small LMs ignore flipped labels, while LLMs can override semantic priors learned during pretraining. Interestingly, Razeghi et al. (2022) demonstrates that in-context learning performance is highly correlated with the prevalence of each instance in the pretraining corpus, showing that models are more accurate on few-shot numerical reasoning on instances whose terms are more frequent.

## 6.2 Selecting Informative Demonstrations

Typically, work on evaluating LLMs in few-shot settings commonly uses randomly sampled examples to compose the in-context prompt (Brown et al., 2020; Zhang et al., 2022a; Chowdhery et al., 2022; Chung et al., 2022; Touvron et al., 2023). Nonetheless, it has been demonstrated that the effectiveness of few-shot performance significantly depends on the selection of in-context examples (Kocielnik et al., 2022; Ye et al., 2023a; Diao et al., 2023; Xu et al., 2023). Consequently, there is ongoing research on generating or selecting the most informative demonstrations, aiming to maximize the downstream few-shot performance.

Some approaches are based on a retrieval component that sources the most relevant examples from a pool. The prompt retriever can be trainable (Rubin et al., 2022) or based on pretrained embeddings (Liu et al., 2022; Agrawal et al., 2022). Gonen et al. (2022) use uncertainty to evaluate the use-

fulness of in-context examples and find that the best performing prompts have low perplexity. Zhang et al. (2022b) formulate example selection for in-context learning as a sequential decision problem and show modest performance improvements by acquiring data with their proposed method based on reinforcement learning. Other previous work, instead of focusing on the part of acquiring data for in-context learning, show that demonstration ordering (Lu et al., 2022) and model calibration (Zhao et al., 2021) are additional properties that influence the few-shot learning performance.

## 6.3 Active Learning for NLP

AL has been extensively studied in various NLP tasks, including machine translation (Miura et al., 2016; Zhao et al., 2020), natural language inference (Snijders et al., 2023), named entity recognition (Shen et al., 2017; Wei et al., 2019), and text classification (Ein-Dor et al., 2020; Margatina et al., 2022; Schröder et al., 2023), among others.

Still, its importance and potential value is on the rise (Zhang et al., 2022c; Rauch et al., 2023), as the current language model pretraining paradigm continues to advance the state-of-the-art (Tamkin et al., 2022). Given the fundamental premise that "not all data is equal" it is reasonable to expect researchers to actively seek the "most informative" data for pretraining or adapting their large language models (LLMs), as well as identifying the most valuable in-context examples for few-shot learning scenarios. Previous work has explored AL for prompt-based finetuning (Köksal et al., 2022), proposing a method based in inter-prompt uncertainty sampling with diversity coupled with the PET architecture (Schick and Schütze, 2021a,b) that outperforms all AL baselines.

## 7 Conclusion

In this study, we have examined the selection of demonstrations, i.e., labeled data that provide examples of solving a task, for in-context learning with LLMs. We formulated the selection process as a *single iteration active learning problem* and evaluated four standard approaches: `uncertainty`, `diversity`, `similarity`, and `random` sampling. Our evaluation involved 15 models of varying size from the GPT and OPT families, encompassing 15 classification tasks and 9 multi-choice tasks. Through extensive experimentation, we have demonstrated that selecting demonstrations that are semantically similar to the test input examples consistently outperforms all other methods by a significant margin across all model families, sizes, and tasks. This corroborates findings of several previous and concurrent studies that explore the properties of "good" in-context examples (Liu et al., 2022; Shi et al., 2022). Interestingly, our findings reveal that uncertainty sampling, although effective in supervised learning, underperforms in the in-context learning paradigm. This highlights the importance of our work in exploring the principles of active learning in the context of few-shot learning.

## Acknowledgements

We would like to thank the anonymous reviewers for their suggestions to improve our work. We also thank Louis Martin, Patrick Lewis, Fabio Petroni and other members of FAIR for their constructive feedback on previous versions of the paper.

## Limitations

**Tasks & Datasets** We acknowledge that even though we experimented with a well established benchmark, the `Crossfit` (Ye et al., 2021) benchmark consisting of 15 classification and 9 multi-choice question answering datasets (Appendix A.1), it might still not be sufficient to ensure that our findings will generalize to any NLP classification or multi-choice application of in-context learning.

**Language** We also acknowledge that all the datasets and models considered in this work are based on the English language alone. This limits generalizability of our findings to other languages.

**Model scale** We investigated in-context learning with actively acquired demonstrations with 15 GPT and OPT models that span 125M to 30B parameters. Even though our experimentation is thorough, our findings might not generalize to larger or smaller transformer-based models, or models based in a different architecture.

**Active learning considerations** We explicitly note in the paper that we do a single active learning iteration, which is different than the common AL loop that consists of multiple iterations. As we explained, because the model-in-the-loop (the LLM) is not updated (no fine-tuning) with new data, performing multiple iterations does not make sense in this context (Figure 2). Still, it would be interesting for future work to explore how we can perform multiple AL iterations while constructing the prompt (i.e., acquiring the demonstrations). The upper bound would be to try all the combinations of a set of labeled data and find the best performing prompt. However, doing this with unlabeled data, in an efficient way, is far from trivial. We refer to Zhang et al. (2022c); Treviso et al. (2023); Margatina and Aletras (2023) for in-depth suggestions for future work in this area.

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

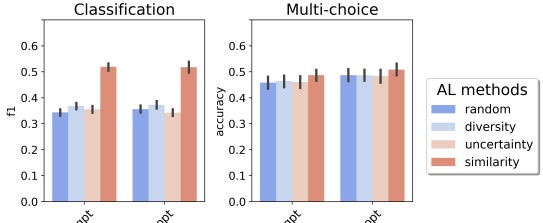

Figure 9: Results per model family.

# A    Experimental Details

## A.1    Tasks & Datasets

Following Min et al. (2022), we evaluate our models in 15 classification and 9 multi-choice tasks taken from the `Crossfit` (Ye et al., 2021) benchmark. Specifically the tasks we evaluate are *poem_sentiment* (Sheng and Uthus, 2020), *glue-wnli* (Wang et al., 2019; Levesque et al., 2012), *climate_fever* (Diggelmann et al., 2020), *glue-rte* (Wang et al., 2019), *superglue-cb* (de Marneffe et al., 2019), *sick* (Minaee et al., 2021), *medical_questions_pairs* (McCreery et al., 2020), *glue-mrpc* (Wang et al., 2019; Dolan and Brockett, 2005), *hate_speech18* (de Gibert et al., 2018), *ethos-national_origin* (Mollas et al., 2022), *ethos-race* (Mollas et al., 2022), *ethos-religion* (Mollas et al., 2022), *tweet_eval-stance_atheism* (Barbieri et al., 2020), *tweet_eval-stance_feminist* (Barbieri et al., 2020) and *quarel* (Tafjord et al., 2019a), *openbookqa*, *qasc* (Khot et al., 2020), *commonsense_qa*, *ai2_arc* (Clark et al., 2018), *codah* (Chen et al., 2019), *superglue-copa* (Gordon et al., 2012), *quartz-with_knowledge* (Tafjord et al., 2019b), *quartz-no_knowledge* (Tafjord et al., 2019b), for classification and multi-choice respectively.

## A.2    Full results

We provide below the full set of results, for each dataset, model and active learning acquisition strategy considered. The dashed line depicts the majority vote baseline.

## A.3    Model Family

We provide the results on few-shot learning with $k$=16 demonstrations per prompt per model family and task type in Figure 9. We observe the same patterns for both GPT and OPT models.

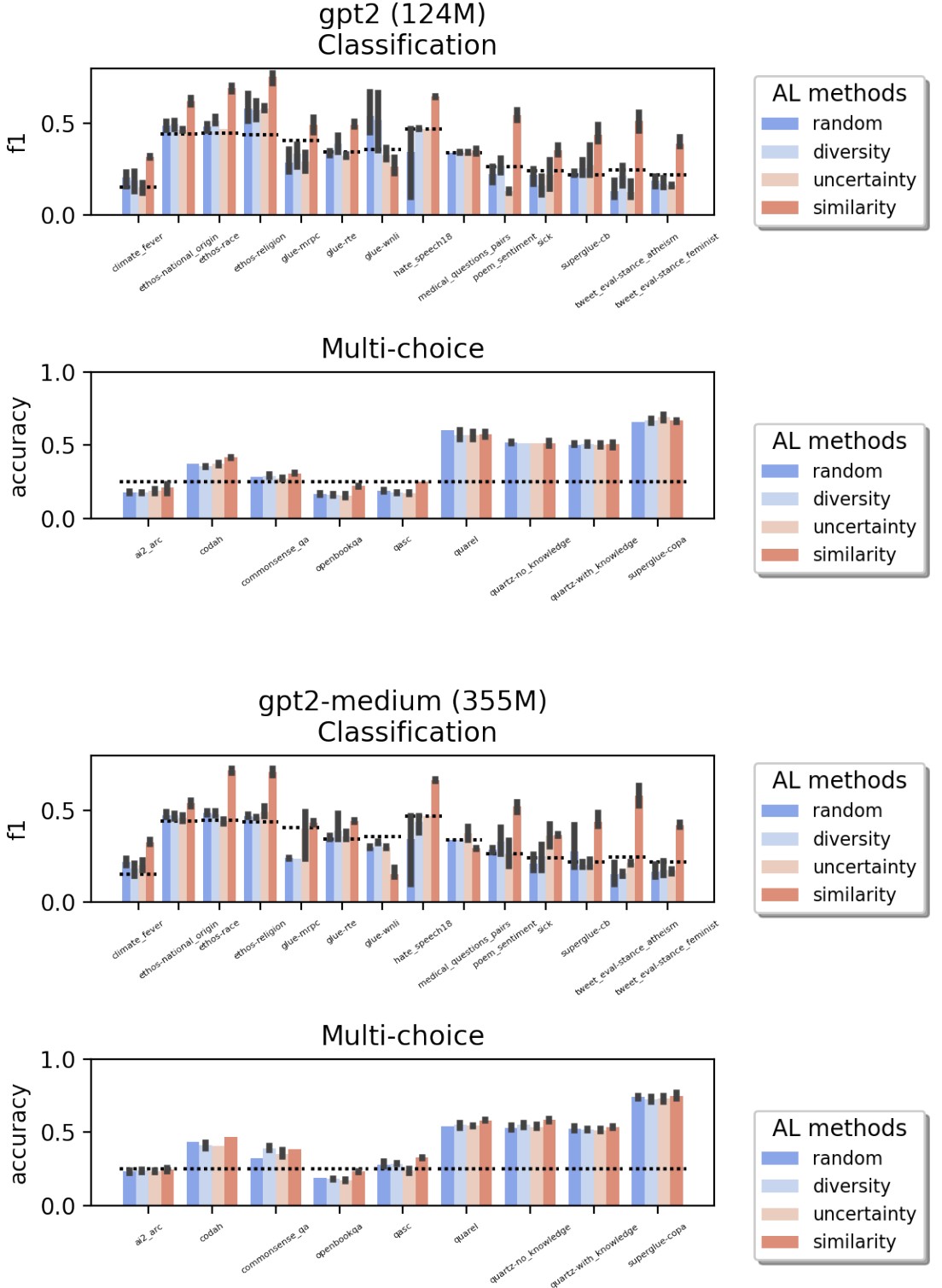

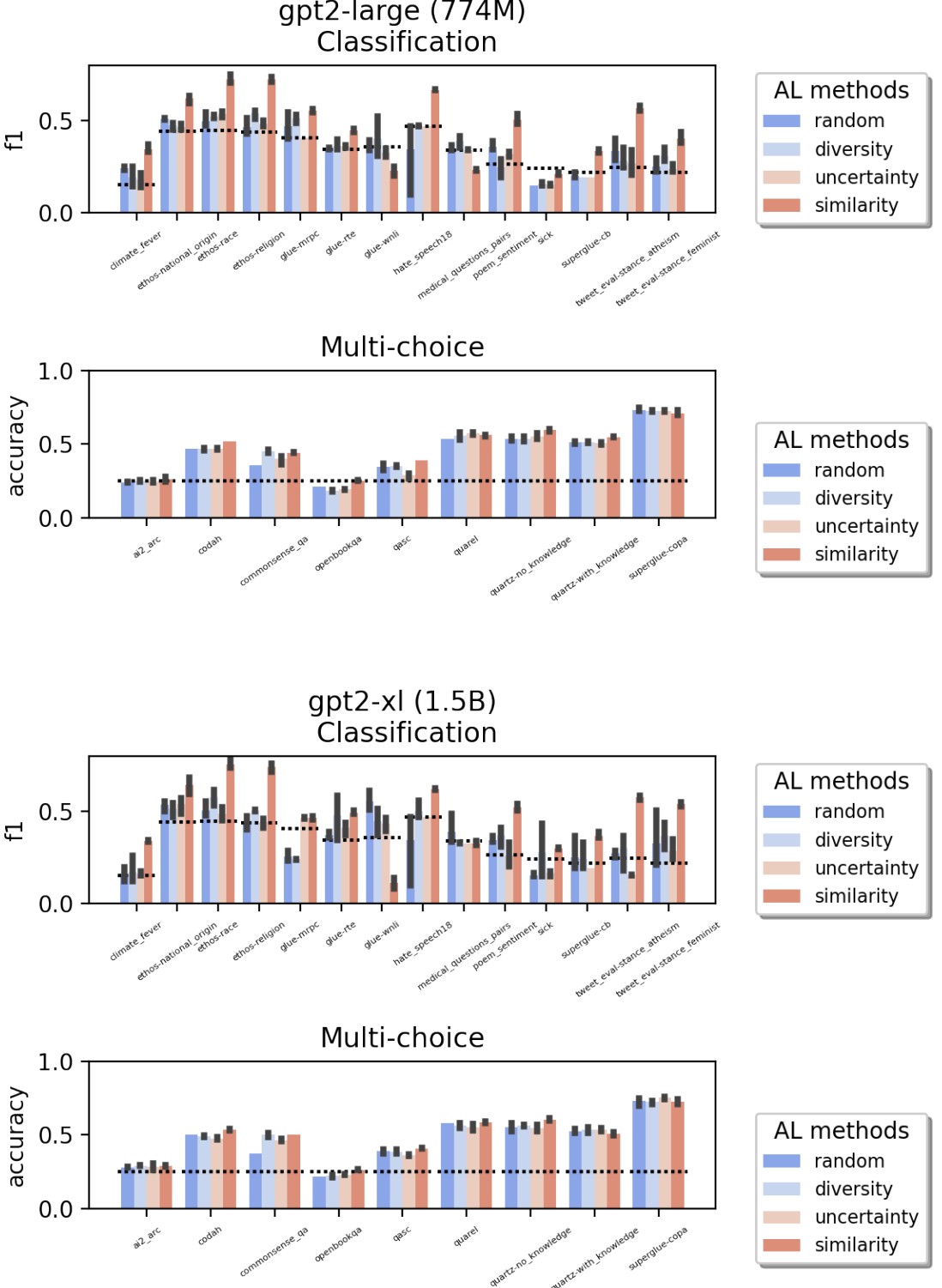

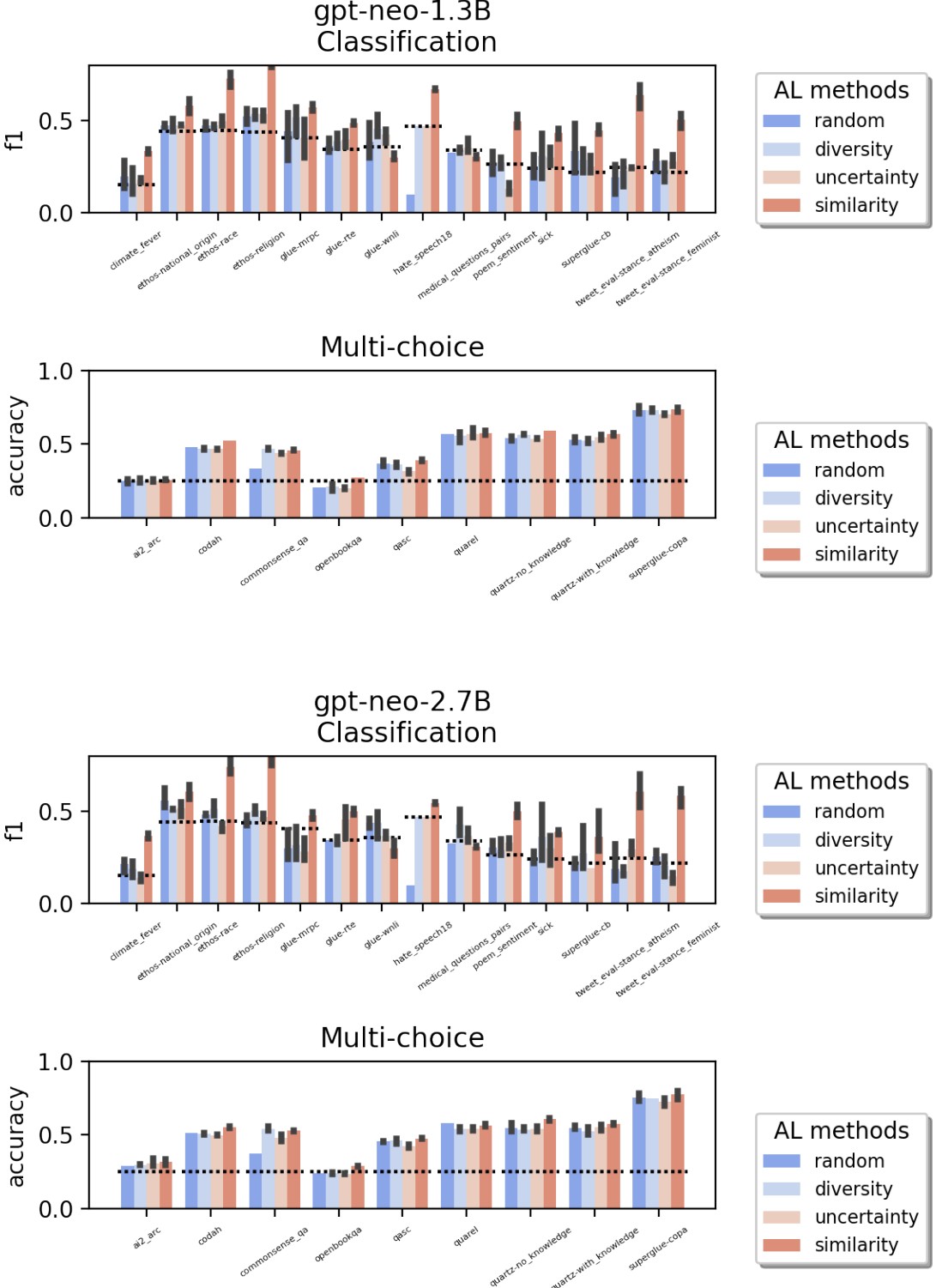

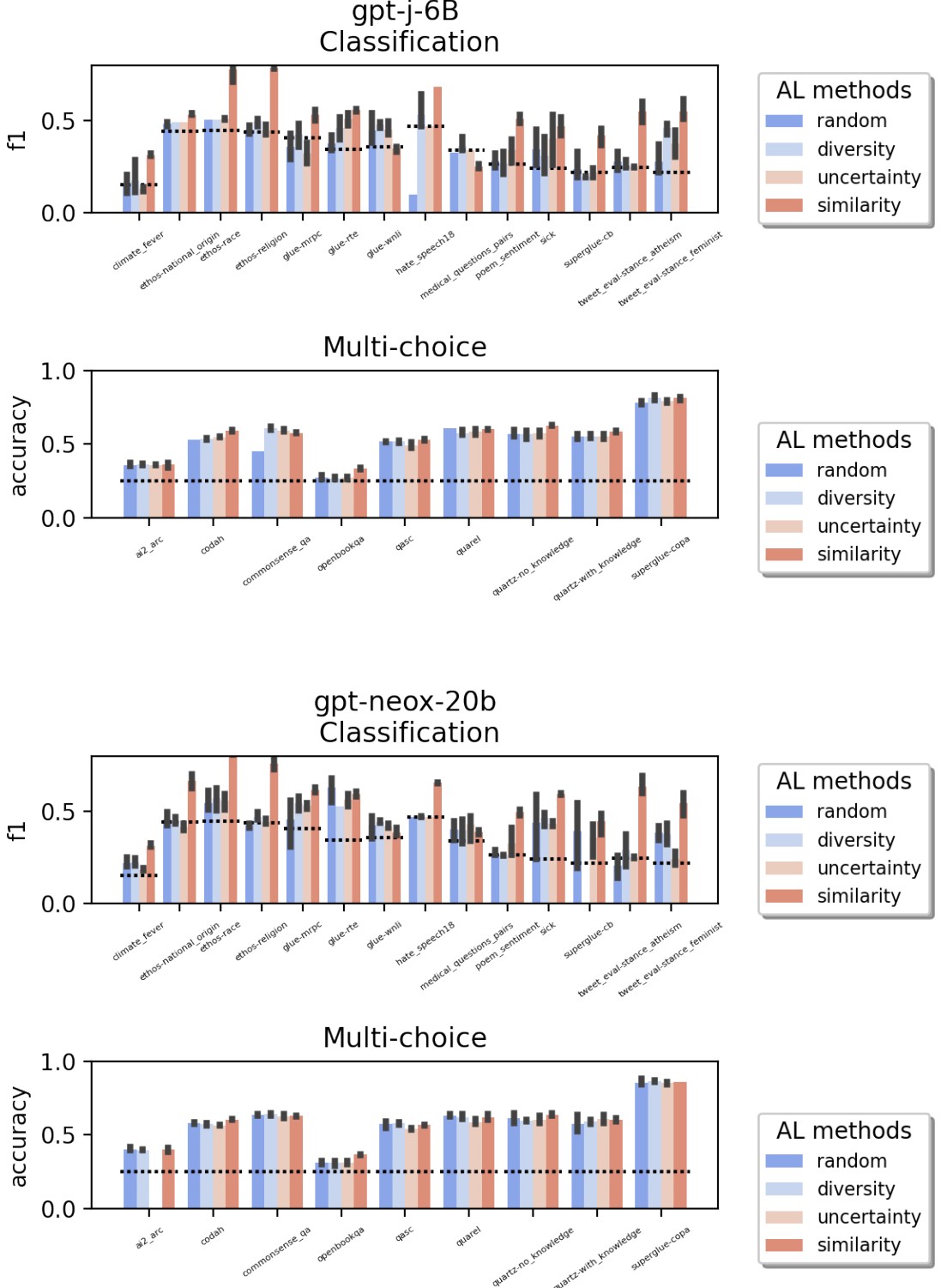

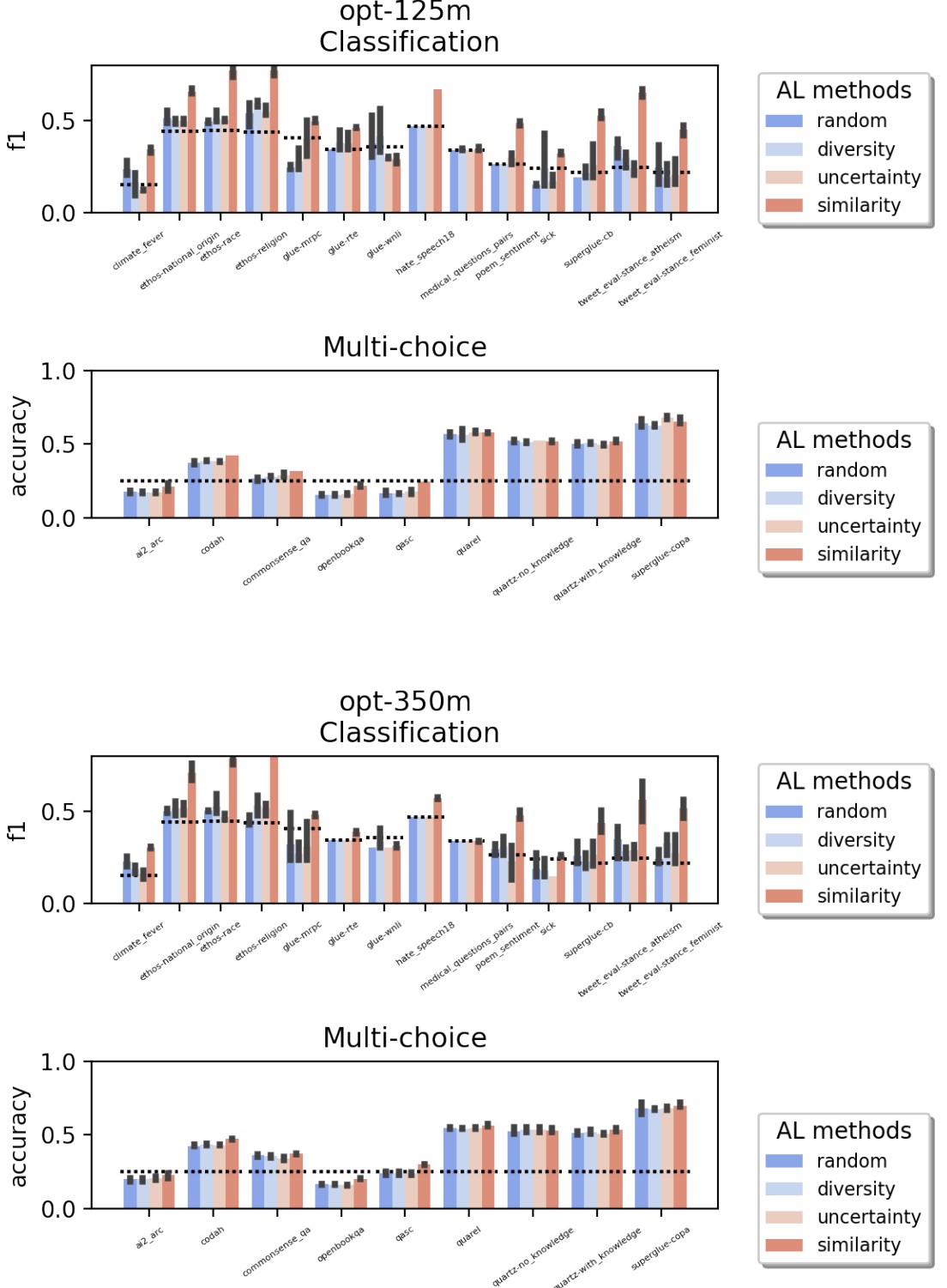

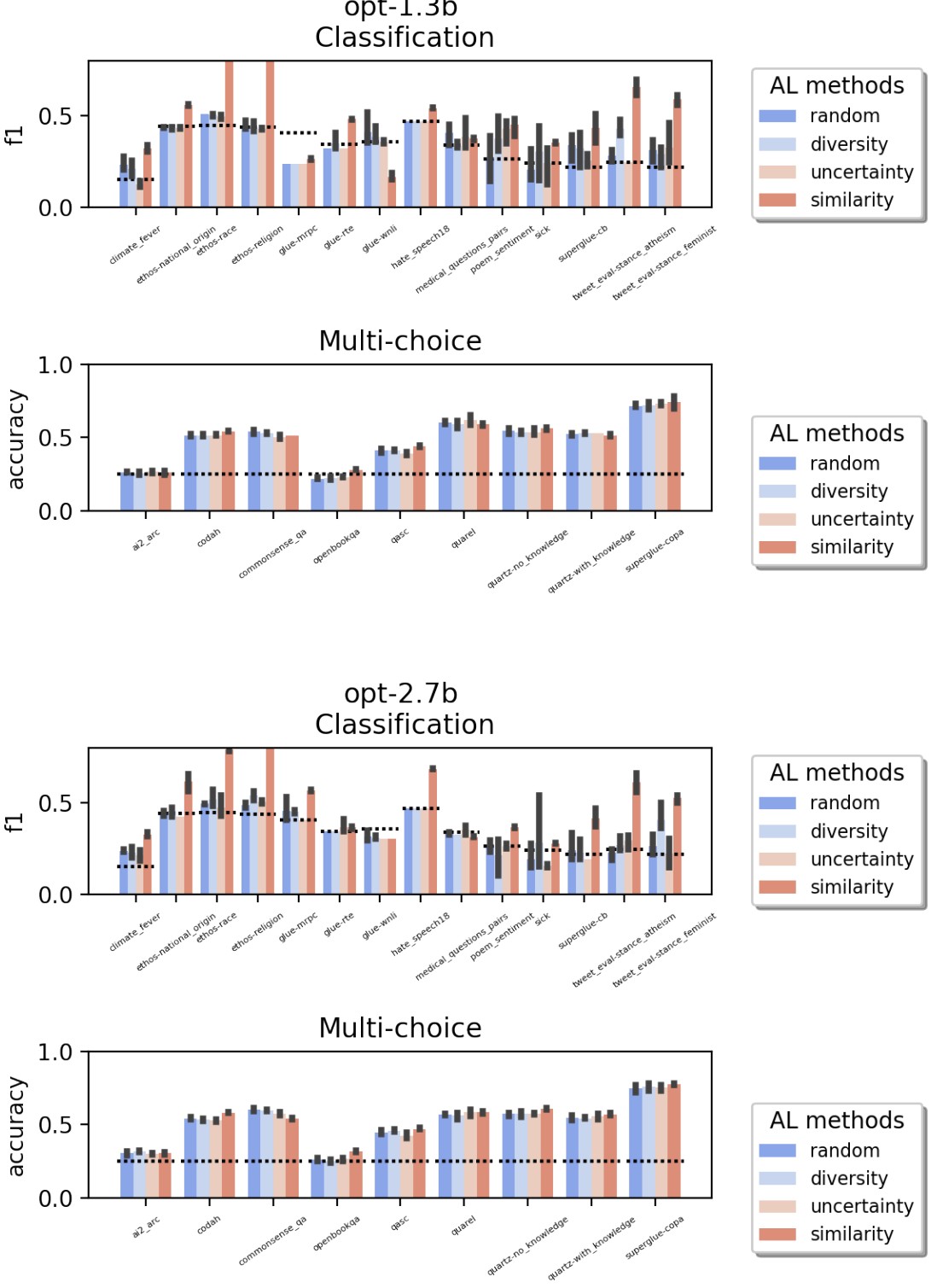

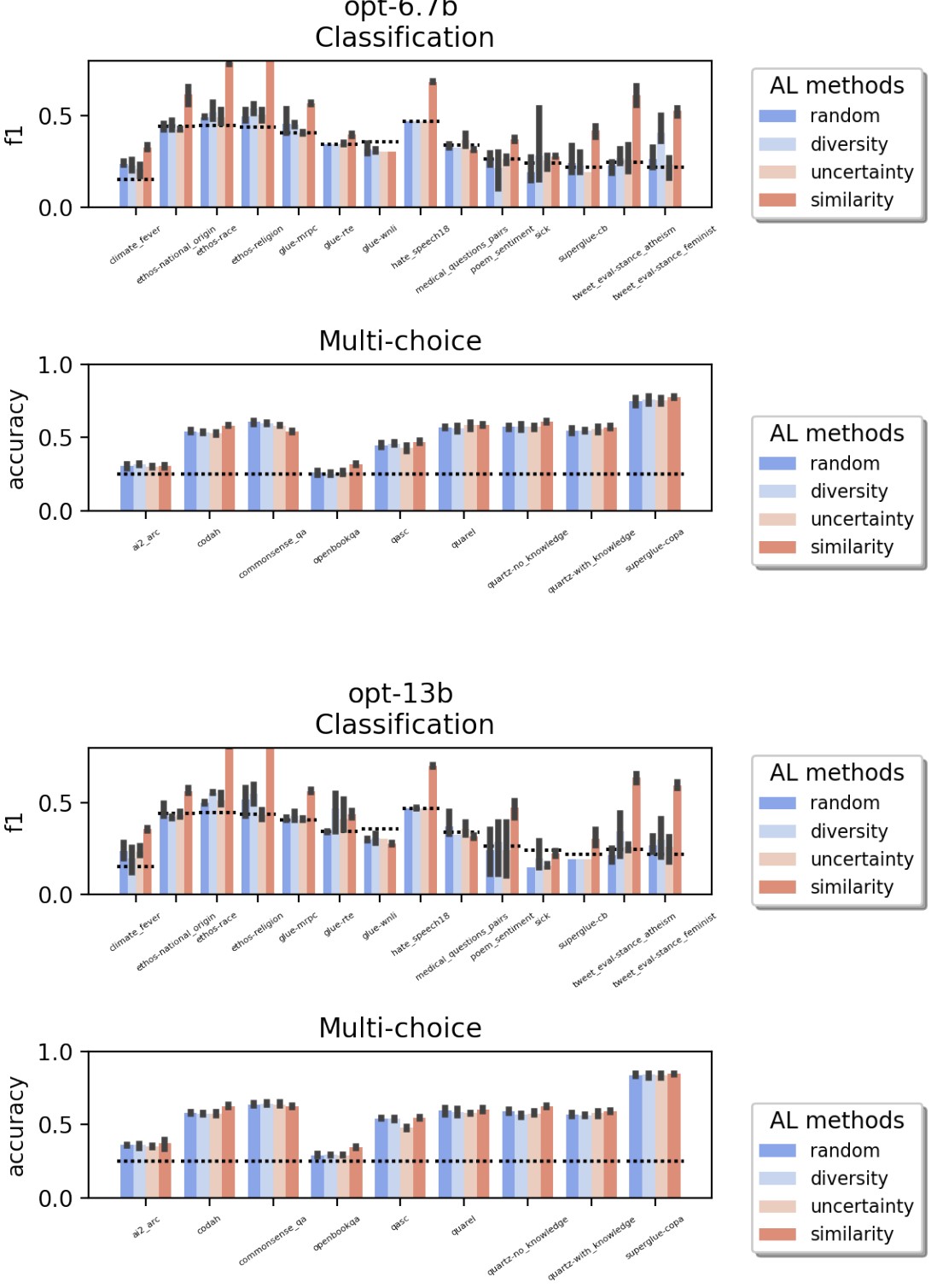

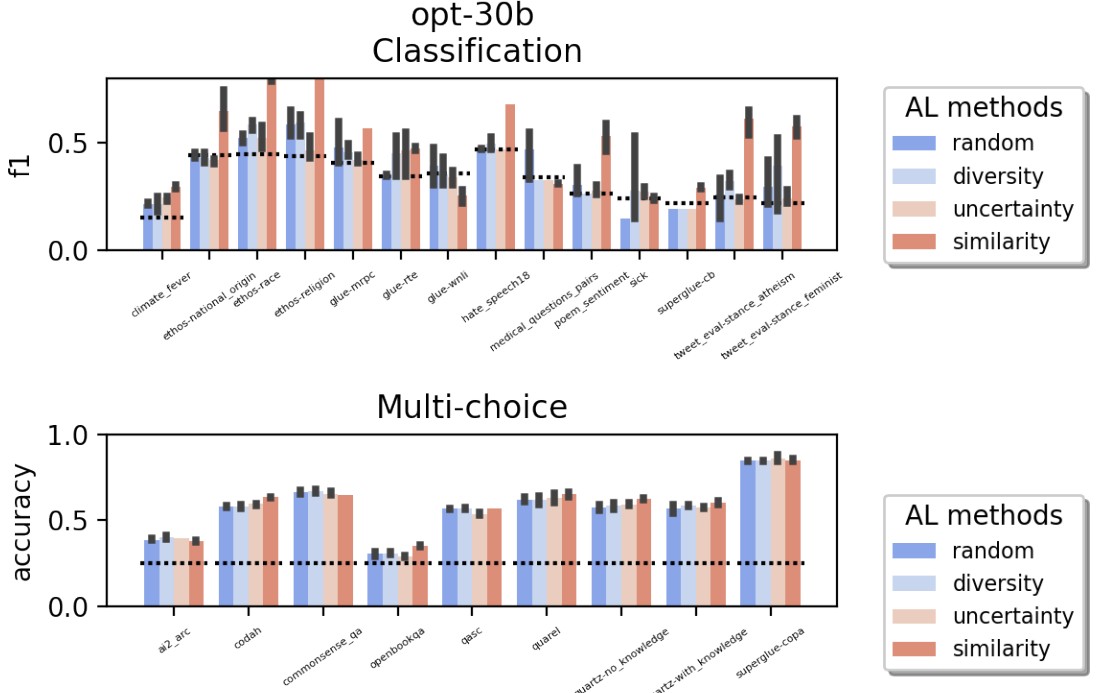