# OpenReview forum: "Active Learning Principles for In-Context Learning with Large Language Models"
_EMNLP/2023/Conference — EMNLP 2023 Findings_

### Official Review · Reviewer_BaZZ · 2023-07-27

**Typos Grammar Style And Presentation Improvements:** 032
**Soundness:** 4

**Excitement:**

4: Strong: This paper deepens the understanding of some phenomenon or lowers the barriers to an existing research direction.

**Paper Topic And Main Contributions:**

The paper studies the effect of using different methods to select exemplars for in-context learning as a function of model family, size, type of task (classification, multiple choice), metric (F1, accuracy), etc. It is a thorough study and it relates to a few works from the literature. The main methods considered are based on active learning (uncertainty-, diversity-, similarity-, and random-based). The general finding is that 1) similarity is very strong; 2) uncertainty is surprisingly not very strong given the norm (albeit for larger model sizes the most uncertain exemplars are quite useful); 3) the results for the order of the methods are metric specific; 4) ground truth labels are actually needed.

The study is thorough, which is a contribution on its own.

**Questions For The Authors:**

line 340: you claim you do not see any specific patterns, but I think there are patterns, at least in the right panel in Figure 4. The left panel is really interesting: it seems that the similarity approach that you study boosts classification performance a lot, and then model size stops improving the performance, relatively speaking. Could you discuss that in a bit more depth? I guess you could say that there is no pattern for the other methods too, i.e. that the other methods (uncertainty, diversity and random) seem to not improve with model size. Interesting, what is the difference between classification and in-context learning here? Maybe the larger the models get the association between the exemplar and its label is better, and it is thus well utilized to make inference about the test sample.

**Reasons To Accept:**

1. The paper is very clearly written, and it attacks an important practical problem in the era of LLMs.

2. The experiments and analysis are well-designed.

3. The results are interesting and could be used as reference to future work.

**Reasons To Reject:**

1. I was quite surprised that given the success of the similarity method, the authors didn't study the role of SentenceBERT embeddings in extracting that similarity. Is it possible that something about SentenceBERT makes the results so good for that method? I would expect the authors to compare SentenceBERT with more modern sentence embedding models, such as SimCSE and DiffCSE. It is particularly important to look at such an ablation study, in order to understand the role of the nature of the embeddings that you use.

**Reproducibility:**

4: Could mostly reproduce the results, but there may be some variation because of sample variance or minor variations in their interpretation of the protocol or method.

**Reviewer Confidence:**

4: Quite sure. I tried to check the important points carefully. It's unlikely, though conceivable, that I missed something that should affect my ratings.

---

> ### Author Rebuttal · Authors · 2023-08-28
>
> We thank Reviewer BaZZ for their comprehensive review of our paper. We are grateful for the valuable feedback and insightful suggestions they have provided. We address each of their points and questions below.
>
> - Role of Sentence Embeddings:
> We thank the reviewer for highlighting the potential influence of SentenceBERT embeddings in driving the strong results obtained with the similarity method. We agree that exploring the role of these embeddings bears profound significance. Within the updated version of our paper, we intend to explicitly present this aspect as a plausible trajectory for subsequent investigation. Such follow-up work could focus on discerning the embeddings that yield optimal advantages for the similarity-based data selection approach and articulating the intrinsic attributes that these embeddings should have. Previous studies and our own research, undoubtedly show that the underpinning factor for high performing in-context learning  lies in the semantic similarity between in-context examples and the test input of the prompt. Consequently, future work could investigate in detail what is the most beneficial and efficient way to perform data selection.
>
> - Analysis of Patterns in Figure 4:
> We appreciate the reviewer's observation of the patterns in Figure 4. Our comment that we do not observe any patterns in L340 was to convey that as the models’ size is increasing, there is not a data selection method that would benefit more than any other method. In other words, the ranking between the data election methods we explored was more or less the same across all model sizes. We will certainly provide a more in-depth discussion in the camera ready version of the paper to address these findings. Furthermore, the reviewer correctly mentioned that few-shot learning with demonstrations selected with the ‘similarity’ method outperforms larger models that use few-shot examples drawn with different approaches in the classification tasks. We believe that this is one of the most interesting findings in our study, showing that it is perhaps better to invest in finding the most useful demonstrations than using larger models that are expensive for inference. We do observe a difference between classification and multi-choice tasks. In the former, we hypothesize that smaller models can harness well-chosen demonstrations to aptly differentiate between classes due to the relatively 'easier' nature of the task. In contrast, multi-choice tasks necessitate more substantial models to achieve satisfactory outcomes. This discrepancy can potentially be attributed to the distinct complexities inherent in each task category. We will delve into this hypothesis and analyze the interplay between model size, demonstration utilization, and the nature of the tasks in greater detail given the extra space in the camera ready upon acceptance.
>
>
> We are thankful for the insightful suggestions and questions that Reviewer BaZZ has brought to our attention. We will incorporate these suggestions into the revised version of the paper to enhance its quality, clarity, and impact.

---

### Official Review · Reviewer_87v7 · 2023-08-01

**Soundness:** 4

**Excitement:**

2: Mediocre: This paper makes marginal contributions (vs non-contemporaneous work), so I would rather not see it in the conference.

**Paper Topic And Main Contributions:**

This paper systematically studies various principles in selecting in-context learning (ICL) examples for few-shot learning, tested on several LLMs of varying sizes and across many classification and multi-choice tasks. The key findings of the paper are 1) semantically similar samples serve as better ICL examples, 2) sampling diverse samples is promising, and 3) perhaps most surprisingly, selecting uncertain samples, while widely used in active learning literature, does not lead to performance improvement for smaller models while the larger models may take advantage of the uncertain demonstrations to a certain extent.

**Reasons To Accept:**

- Given the popularity of few-shot learning and in-context learning in LLMs, selecting the ICL examples has become a hot area of research. This paper presents a timely and systematic analysis of this topic, and its findings will likely impact future works working on similar areas.

- The analysis is largely thorough and extensive, considering the number of tasks and models used. This improves the reader's confidence in the generalizability of the findings.

**Reasons To Reject:**

While the authors have tried to argue the novelty of their paper in treating the ICL example selection problem as an active learning example and claimed that they are the first to do so, I have not fully grasped any substantive difference from previous works in terms of execution of the algorithm, and the claimed difference seems to lie fully in narrative and rhetoric -- I am therefore requesting the authors to clarify this crucial point in the author rebuttal in how the proposed casting of the problem into an active learning one *substantively* differentiate the paper from the previous works on similar topics. Specifically, my concerns are:
  - Finding 1, as I listed in *Paper Topic And Main Contributions*, is largely known from the widely-cited KATE paper the authors also cited in their paper. While it is nice that authors show that retrieving similar queries as ICL examples works the best across a wide number of model and task combinations, I wonder in what ways this paper furthers or deepens understanding of this phenomenon apart from confirming the KATE paper?
  - While there might not be previous work explicitly presenting Finding 2 in the exactly same setup, many papers have hinted at this point, and an example is AutoCoT which concludes that retrieving most similar examples does not work when the demonstrations are imperfect and maximising diversity improves performance. Again, while this paper confirms that such benefit exists even in the few-shot setup, which is expected, I have not fully grasped the implication beyond that, similar to point 1.
  - Finding 3 is perhaps the most surprising and most related to the active learning paradigm the authors have referred to. However, the paper neither provides convincing explanations nor practical next steps on how to act upon this (although I agree this is an interesting finding).

Overall, while this paper presents very solid empirical evidence and can be potentially useful as a technical report, it seems to me that it is mostly confirming known phenomena and has provided limited further insights. A possible way forward, in my opinion, is given that the authors have confirmed the superiority of similarity and diversity (and the general inferiority of uncertainty). Is there an optimal way to combine these criteria into a novel selection procedure that balances these principles, or, alternatively, is the learning-based method such as [1] implicitly performing retrieval based on one or more criteria mentioned in the paper? These are just some non-exhaustive suggestions that the authors may possibly consider.


### References

[1] Rubin, O., Herzig, J., & Berant, J. (2021). Learning to retrieve prompts for in-context learning. arXiv preprint arXiv:2112.08633.


***

After rebuttal: I thank the authors for their detailed feedback. I agree with the authors' claim on the extensiveness and soundness of the experiments, and I remain convinced that this paper is sound, can potentially be an influential source for future references, and therefore spur further research in this direction -- I have therefore kept my high soundness score.

Regarding the excitement assessment, I have the impression that the author feedback largely confirmed my initial assessment. While the experiments are indeed thorough across a wide range of tasks and model choices, since the excitement score specifically asks for aspects of novelty and interestingness, I tend to keep my original score given that most analyses are re-confirmations and extensions of known results. There are indeed some new results, such as the effect of model size and the role uncertainty plays, but like I mentioned in the original review, I still think the arguments, especially regarding their practical uses, are rather underdeveloped. Although the authors made several promises of future exploration in response to my raised points, unfortunately, such extensions are yet to be done. Please be aware that I am not as negative as the score description would otherwise suggest literally, but I still regard the novelty aspect as a major, pending concern.

**Reproducibility:**

4: Could mostly reproduce the results, but there may be some variation because of sample variance or minor variations in their interpretation of the protocol or method.

**Reviewer Confidence:**

4: Quite sure. I tried to check the important points carefully. It's unlikely, though conceivable, that I missed something that should affect my ratings.

---

> ### Author Rebuttal · Authors · 2023-08-28
>
> We thank Reviewer 87v7 for their detailed and insightful review of our paper. We appreciate the reviewer’s comments that our findings will likely impact future studies on this area, while our extensive analysis improves the reader's confidence in the generalizability of the findings.
>
>  We respond to each of their comments and suggestions below.
>
> - Clarification on Novelty:
> We appreciate the reviewer's comment regarding the novelty of treating the in-context learning (ICL) example selection as an active learning (AL) problem. Throughout our paper, we have refrained from claiming the introduction of a novel AL algorithm specifically tailored for the ICL example selection task. Instead, we recognize the parallels between data selection in ICL and the AL paradigm. Our aim was to conduct an extensive empirical study that explores how established algorithms from the AL domain could be adapted and applied to the ICL framework. We have incorporated a comprehensive compilation of related papers from both the AL and ICL fields, as expected, aiming to offer researchers a go-to guide as they address the problem of few-shot learning data selection via the AL lens. Our intention is to present an empirical study that offers a comprehensive analysis of the strengths and limitations of existing data selection methodologies inspired by AL algorithms (we chose the relevant track “Language Modeling and Analysis of Language Models” for submission accordingly). Conducting an exhaustive series of experiments across diverse models, ranging in size and family, as well as multiple tasks and datasets, constituted a pivotal facet of our study. We acknowledge that the results we obtained corroborate in part previous findings, and we have clearly acknowledged this alignment in our paper. We do not consider this problematic, given that we had a different starting point and a more thorough experimental setup. Surprisingly, we showed that smaller models with demonstrations drawn with the similarity method can outperform larger ones that use different in-context selection methods (Figure 4, OPT models in classification tasks). This is an impactful finding that has not been presented before in previous work. Nevertheless, we believe that the true novelty of our paper resides predominantly within the thorough analysis in Section 5. Our analysis explores the properties of "good" in-context examples and lays the foundation for future research to devise algorithms grounded in these properties. Sections 5.3 and 5.4 distinctly encapsulate our exploration of this pivotal research question. The implications we unveil, particularly those pertaining to a model's capacity to leverage uncertainty and the role of model scale (Section 5.4), resonate as intriguing, influential, and indeed novel insights. We deeply value the reviewer's feedback, and we are committed to elucidating our contributions more explicitly in the revised manuscript.
>
> - Addressing Known Phenomena:
> As previously outlined, we acknowledge that while some of our findings may align with previous work, the application of these principles to the specific context of analysing AL inspired methods for few-shot learning and the systematic evaluation across models and tasks still provide valuable insights. Specifically regarding the highly cited KATE paper, our investigation surpasses its scope by exploring the impact of selecting in-context examples semantically similar to the test input across a comprehensive array of 15 models and 24 datasets encompassing multiple tasks. In contrast, KATE's purview was confined to GPT-3, limited to three datasets spanning sentiment analysis, table-to-text generation, and question answering. Our extended experimental setup not only reaffirms the utility of selecting such demonstrations but also substantiates the generalizability of this conclusion across diverse tasks and models. Furthermore, the importance of our work lies in our exploration of the similarity method across a vast spectrum of model sizes, spanning from a few million parameters to tens of billions. This robust spectrum substantiates the rather surprising finding that prioritizing the similarity-based example selection methodology can outweigh the relative influence of model size in the classification. This hints the prospect that investing in high-quality demonstrations could be more beneficial than deploying resource-intensive large-scale models, which often entail considerable operational costs. To deepen our contribution, we will include a more extensive discussion on the implications of these findings, separating us from prior work. We will better highlight all these points given the extra space in the camera ready upon acceptance.
>
> - Combining Criteria and Further Directions:
> We appreciate the reviewer's suggestions for exploring the combination of similarity, diversity, and uncertainty in a novel selection procedure. We agree that this avenue holds promise for advancing the data selection problem we explore. In the revised version upon acceptance, we will discuss the potential benefits and challenges of such combinations, acknowledging that a balanced approach might yield even better results. We will provide further discussion on how this finding could shape future research directions, potentially leading to strategies that, for instance, exploit uncertainty more effectively for larger models or specific tasks.
>
> We are grateful for the reviewer's suggestions to enhance the clarity, significance, and applicability of our work. We believe that their concerns can be easily addressed in the camera ready version of the paper  by providing a more clear presentation of our contributions. We acknowledge that the revisions proposed will elevate the quality of our paper and its potential impact on the community.

---

### Official Review · Reviewer_RZBu · 2023-08-03

**Typos Grammar Style And Presentation Improvements:** Line 032
**Soundness:** 3

**Excitement:**

3: Ambivalent: It has merits (e.g., it reports state-of-the-art results, the idea is nice), but there are key weaknesses (e.g., it describes incremental work), and it can significantly benefit from another round of revision. However, I won't object to accepting it if my co-reviewers champion it.

**Missing References:**

I recommend writing Srivastava et al instead of listing all the author names.

**Paper Topic And Main Contributions:**

This paper studies several active learning methods that aim at selecting a subset of demonstrations from a pool to then use them as prompts to large language models. The authors analyze four simple approaches based on uncertainty, diversity, similarity, and random sampling. The experimental results on GPT and OPT models with several sizes suggest that the similarity between the selected demonstrations and the test example makes for the best-performing prompt.

**Questions For The Authors:**

- How would you expect the ranking of the different methods to change based on the choice of $k$?
- Overall, I find it unclear what are the actual contributions of the work since most findings have already been observed in previous work. Can you please clarify this?

**Reasons To Accept:**

- The paper is clear and well-written.
- The paper tackles the important and timely problem of improving the prompting of pre-trained language models such that no fine-tuning is needed.

**Reasons To Reject:**

- The diversity and similarity measures use different clustering methods (k-means vs knn) which might cloud the proper assessment of each attribute. Instead of using $k$NN for the similarity measure, you can use SBERT embedding clustering and pick $k$ random points in the same cluster as the test point (or the closest cluster if there aren't $k$ points in the test point cluster).
- The contradictory conclusions when using different evaluation metrics (F1 vs accuracy) are rather alarming and suggest that more work needs to be done to reach a solid conclusion. Particularly, this might suggest that a combination of similar and diverse examples in a given prompt might achieve the best overall performance.
- The effect of the number of demonstrations $k$ is also unclear; exploring how different $k$ values impact the studied AL methods (and their relative ranking) seems pertinent.

**Reproducibility:**

4: Could mostly reproduce the results, but there may be some variation because of sample variance or minor variations in their interpretation of the protocol or method.

**Reviewer Confidence:**

2: Willing to defend my evaluation, but it is fairly likely that I missed some details, didn't understand some central points, or can't be sure about the novelty of the work.

---

> ### Author Rebuttal · Authors · 2023-08-28
>
> We thank Reviewer RZBu for their thoughtful and constructive review of our paper. We appreciate that the reviewer considers our paper clear and the subject of our study to be of high importance. We address each of their comments below.
>
> - Choice of Clustering Methods:
> We acknowledge the comment on using different clustering methods for diversity and similarity methods, potentially clouding their proper assessment. Still, we believe that the fair comparison between diversity and similarity methods  is justified by the use of the same representations (SBERT embeddings). Furthermore, given that k-means is an unsupervised clustering technique, it is non-trivial how it could be applied to the similarity selection method. We opted to use algorithms proposed in prior work off-the-shelf for a fair comparison and to avoid confusion. We will clarify this point in the revised version of the paper.
>
> - Contradictory Conclusions:
> We appreciate the reviewer's observation regarding contradictory conclusions based on different evaluation metrics. However, this is a common issue with in-context learning examples that pertains across the literature [1,2]. Our intention was to better highlight this issue rather than hiding it by including the results for both evaluation metrics in the main body of the paper.  We will make sure to clarify this part more in the camera ready version of the paper. We agree that combining both similar and diverse examples in a prompt is insightful and an interesting avenue for future work. In this paper we opted to focus on an extensive empirical study  of general active learning (AL) inspired data selection methods instead of trying to propose a specific algorithm. We hope that the findings and analysis of our paper will inspire future work on the impact of similarity, diversity and uncertainty, and explore the synergies between them. We will make sure to propose explicit avenues for further research in the updated version of the paper.
>
> - Impact of Number of Demonstrations & Method Ranking:
> The reviewer's point about the impact of the number of demonstrations on the studied AL methods and their relative ranking is well-taken. We had explored different numbers of k (1,4,8,16) and did not find any different patterns (i.e., different selection methods performed similarly, independently of the choice of k). This is why we chose to perform the bulk of the experiments with k=16, as in prior work [1,2]. We will include a short analysis that explores different numbers of demonstrations and their effects on method performance in the final version of the paper.
>
> - Clarification of Contributions & Soundness:
> We regret any ambiguity in conveying the contributions of our work. While some findings have been observed in previous studies, our paper offers a systematic comparison of different AL-inspired methods specifically in the context of prompting pre-trained language models. We have conducted a thorough study of 15 language models and 26 datasets, comparing 4 representative data selection methods. We believe that the soundness of our experimental work is solid, which is also mentioned by reviewers 87v7 and BaZZ (who both gave us a score of 4). Furthermore, our experiments illustrate that prioritizing the application of the similarity method in demonstration selection can outweigh the significance of model size in classification tasks. For instance, we observe in Figure 4 that OPT models from 125M to 30B parameters perform almost equally when using the similarity method. This surprising finding suggests that investing in superior-quality demonstrations could yield more favorable outcomes compared to the utilization of larger, cost-intensive models, which leads to favorable implications for model efficiency in general. Additionally, our investigation into the interplay between the top and bottom candidate examples from the pool in terms of uncertainty and similarity (Section 5), across different model families and number of parameters, contribute novel insights to the field. We will revise the paper upon acceptance to better emphasize these contributions.
>
> We are committed to addressing all concerns and suggestions raised by the reviewer in order to improve our paper. We believe that the revisions proposed will strengthen the overall contribution and merit of the work.
>
>
> [1] Rethinking the role of demonstrations: What makes in-context learning work? Min et al., 2022.
> [2] Ground-truth labels matter: A deeper look into input-label demonstrations. Yoo et al., 2022.

---

### Meta-Review · Area_Chair_aLBM · 2023-09-18

**Recommendation:** 4

**Metareview:**

This paper studies active learning methods for selecting demonstrations to prompt large language models. Four simple approaches are analyzed: uncertainty, diversity, similarity, and random sampling. Results on GPT and OPT models suggest that similarity to the test example is best.

Pros:
- The problem that the paper is solving is important and relevant to the community.
- Thorough and extensive analysis of the topic
- The findings of the evaluations in the paper are relevant to the practitioners.

Cons:
- Novelty of the findings in the paper. Some of the findings are confirming already known results.
- Some of the findings are contradictory across different metrics.

---

### Decision · Program_Chairs · 2023-10-07

**Decision:**

Accept-Findings

**Comment:**

This paper studies active learning methods for selecting demonstrations to prompt large language models. Four simple approaches are analyzed: uncertainty, diversity, similarity, and random sampling. Results on GPT and OPT models suggest that similarity to the test example is best.

Pros:
- The problem that the paper is solving is important and relevant to the community.
- Thorough and extensive analysis of the topic
- The findings of the evaluations in the paper are relevant to the practitioners.

Cons:
- Novelty of the findings in the paper. Some of the findings are confirming already known results.
- Some of the findings are contradictory across different metrics.